# Uncertainty and non-stationarity of empirical streamflow sensitivities

Sebastian Gnann<sup>1</sup>, Bailey J. Anderson<sup>2,3,4</sup>, Markus Weiler<sup>1</sup>

<sup>1</sup>Faculty of Environment and Natural Resources, University of Freiburg, Freiburg, 79098, Germany

Correspondence to: Sebastian Gnann (sebastian.gnann@hydrologie.uni-freiburg.de)

Abstract. The sensitivity of streamflow to changes in driving variables such as precipitation and potential evaporation is a key signature of catchment behaviour. Due to increasing interest in climate change impacts, streamflow sensitivities derived from observations have become a widely used metric for catchment characterization, model evaluation, and observation-constrained projections. However, there remain open questions regarding the robustness and stationarity of empirically-derived sensitivities. In this paper, we revisit theoretical and empirical approaches to estimate streamflow sensitivities to precipitation and potential evaporation. First, we compare different estimation methods – primarily based on linear regression – using a synthetic dataset for which the sensitivities are known. Second, we extend this comparison and use two methods selected based on the previous analysis to estimate sensitivities for >1000 near-natural catchments. Third, we investigate how sensitivities change over time due to changes in the ratio between potential evaporation and precipitation (i.e., aridity index). Our results confirm that multiple regression is preferable to single regression, but that in presence of noise and correlation between precipitation and potential evaporation, even multiple regression methods can lead to high uncertainty, especially for potential evaporation. When analysing real catchments, sensitivity to precipitation is estimated consistently across methods, while sensitivity to potential evaporation is highly uncertain and often yields unrealistic values. Further, as the aridity index increases over time – a trend found in observational data – sensitivities decrease (by 22-70% over 50 years) and are thus non-stationary. These results should urge caution in the use of empirical streamflow sensitivities and call for further investigation.

<sup>&</sup>lt;sup>2</sup>WSL Institute for Snow and Avalanche Research SLF, Davos Dorf, Switzerland

<sup>&</sup>lt;sup>3</sup>Institute for Atmospheric and Climate Science, ETH Zurich, Zurich, Switzerland

<sup>&</sup>lt;sup>4</sup>Climate Change, Extremes and Natural Hazards in Alpine Regions Research Center CERC, Davos Dorf, Switzerland

#### 5 1 Introduction




In response to ongoing interest in climate change impacts, the sensitivity of streamflow to changes in climatic drivers has received considerable attention in recent years (e.g., Andréassian et al., 2016; Lehner et al., 2019; Zhang et al., 2023). In particular, sensitivities to precipitation, temperature, and potential evaporation are commonly used, though the concept can be extended to other driving variables that influence streamflow, such as storage (e.g., Berghuijs et al., 2016; de Lavenne et al., 2022; Weiler et al., 2025; Zhang et al., 2023) or land cover changes (e.g., Anderson et al., 2022; Roderick & Farquhar, 2011; Steinschneider et al., 2013).

Sensitivity is defined as the change in streamflow per unit change in a certain driving variable. This can be expressed in absolute terms (e.g., by how many mm streamflow changes per mm change in precipitation) or in relative terms (e.g., by how many %-points streamflow changes per 1% change in precipitation), the latter often being called elasticity (Andréassian et al., 2016; Sankarasubramanian et al., 2001). We will define both terms more formally later on.

One of the earliest studies explicitly mentioning the "sensitivity of water resource systems to climate variations" was by Němec & Schaake (1982), who used the Sacramento Soil Moisture Accounting model to estimate sensitivity of streamflow to changes in precipitation and potential evaporation for two contrasting basins (an arid and a humid one). This approach comprises one typical category of sensitivity studies, namely perturbation experiments using (process-based) simulation models of varying complexity (e.g., Němec & Schaake, 1982; Nijssen et al., 2001; Schaake, 1990). Another category are simple Budyko-type water balance models (Budyko, 1976), which provide analytical solutions describing the catchment water balance, typically at the climatological time scale. These models calculate streamflow and (complementary to this) actual evaporation as a function of precipitation, potential evaporation, and sometimes other factors (e.g., related to vegetation or land use), and can be used to obtain sensitivity estimates by taking partial derivates to the different driving variables (Andréassian et al., 2016; Dooge, 1992; Harman et al., 2011; Roderick & Farquhar, 2011; Sankarasubramanian et al., 2001). Lastly, empirical approaches have been used to derive sensitivities directly from observational data by means of data-based estimators or linear regression approaches (Andréassian et al., 2016; Chiew, 2006; Sankarasubramanian et al., 2001). Empirical approaches tend to invoke fewer assumptions and are observation-based, thus providing a benchmark for simple analytical models or complex process-based models. However, they are associated with uncertainty due to methodological choices and are influenced by the quality of the underlying observational data.

When studying streamflow sensitivity, it is important to consider the time scale under investigation. Many studies focus on long-term average fluxes (i.e., the climatological time scale), as this is of direct interest when studying how catchments respond to changes in climatic conditions. To empirically derive long-term sensitivities, it is common to aggregate time series of streamflow and its driving variables into annual values and then make use of year-to-year variability to approximate how catchments respond to long-term changes (Andréassian et al., 2016; Sankarasubramanian et al., 2001). Alternatively, it has



been suggested to estimate sensitivities using multi-year averages (e.g., decadal averages; Zhang et al., 2022), though this requires long enough time series to be able to use non-overlapping blocks. In addition, it is worth noting that sensitivities can be assessed in many different ways – for example, by examining annual sensitivities in relation to sub-annual variations (e.g., warm and cold seasons; Ban et al., 2020), by considering different parts of the streamflow regime (e.g., multiple streamflow percentiles; Anderson et al., 2024), or by focusing on different time scales altogether (e.g., weekly sensitivities; Weiler et al., 2025).

Independent of how they are estimated, sensitivities (or elasticities) can be used in a variety of ways. They provide a model-derived summary metric describing how a hydrological system responds to change (Němec & Schaake, 1982; Schaake, 1990). They can be used for direct projections of climate change effects using simple sensitivity models (Roderick & Farquhar, 2011) or observation-based regression approaches (Zhang et al., 2023). They can be used for model evaluation (Wagener et al., 2022) and for constraining model ensembles (Lehner et al., 2019). Lastly, they generally serve as means for catchment characterization (Addor et al., 2018) and thus are frequently used in catchment classification studies (e.g., Almagro et al., 2024; Sawicz et al., 2011).

Due to their widespread use in the context of climate change impact research (e.g., Lehner et al., 2019; Zhang et al., 2023), it is important to have a clear understanding of both the concept of sensitivity and the methods used to estimate it. Recent studies often reported unrealistic (i.e., positive) values for empirically estimated sensitivities to potential evaporation (Andréassian et al., 2025; Awasthi et al., 2024; Xiao et al., 2020), as well as non-stationarities in sensitivities and/or rainfall-streamflow relationships (Anderson et al., 2025; Fu et al., 2007; Matanó et al., 2025; Peterson et al., 2021; Saft et al., 2015; Tang et al., 2019). This warrants a closer look at existing methods used to empirically estimate sensitivities. The central aim of this paper is therefore to test the robustness of different empirical estimation methods and then to investigate potential non-stationarity of sensitivities. To do so, we (1) test different approaches based on an analytical model for which sensitivities are known, (2) apply two selected methods to a large sample of near-natural catchments, and (3) explore how empirical sensitivities change over time using long streamflow time series.

95

100

## 80 2 Theory and methods

In the following, we first present formal definitions of streamflow sensitivity and elasticity (Section 2.1) and introduce the different empirical estimation methods tested in this paper (Section 2.2). These methods are then both applied to a synthetic dataset based on an analytical model (described in Section 2.3) and to observational data (data sources described in Section 3). Finally, we investigate how sensitivities change over time, both theoretically and using observational data (Section 2.4).

## 85 2.1 Definitions of sensitivity and elasticity

Sensitivity s is defined as the absolute change in streamflow  $\Delta Q$  per absolute change in a certain driving variable, here precipitation  $\Delta P$  and potential evaporation  $\Delta E_p$ .  $\Delta Q$  indicates the deviations at each time step t (here usually years) from the long-term average  $\bar{Q}$ ; for example, for streamflow we get  $\Delta Q = Q(t) - \bar{Q}$ . The  $\Delta$  notation is used here due to our focus on empirical sensitivities. When studying sensitivities using analytical equations, this becomes the (partial) derivative  $\partial$ .

$$s_P = \frac{\Delta Q}{\Delta P} \tag{1}$$

$$s_{Ep} = \frac{\Delta Q}{\Delta E_p} \tag{2}$$

Elasticity e is closely related but focuses on percentage changes, so that both streamflow and its driving variables are normalised by their respective means  $\bar{Q}$ ,  $\bar{P}$ , and  $\bar{E}_p$ .

$$e_P = \frac{\Delta Q}{\overline{O}} / \frac{\Delta P}{\overline{P}} = s_P \frac{\overline{P}}{\overline{O}} \tag{3}$$

$$e_{Ep} = \frac{\Delta Q}{\overline{Q}} / \frac{\Delta E_p}{\overline{E_p}} = s_{Ep} \frac{\overline{E_p}}{\overline{Q}} \tag{4}$$

While both sensitivity and elasticity are dimensionless, they are often expressed in mm/mm or %/%, respectively.

An interesting feature of elasticities is the hypothesis that there exists a so-called complementary relationship between  $e_P$  and  $e_{Ep}$  (which is embedded in Eq. 10 in Dooge, 1992, even though he did not explicitly mention that term). This complementary relationship states that the elasticities to P and  $E_p$  should sum up to 1, which was shown to be a characteristic of any Budykotype equation where Q is solely determined by P and  $E_p$  (Zhou et al., 2015), but may not necessarily be true for real catchments.

$$e_P + e_{Ep} = 1 \tag{5}$$

# 2.2 Estimation methods of empirical sensitivities

We test several methods to calculate sensitivities empirically, all listed in Table 1. In addition to the commonly used non-parametric method (Eq. 6; Sankarasubramanian et al., 2001), we use different regression-based methods (Andréassian et al., 2016). These include single linear regression (Eq. 7), multiple linear regression using absolute annual values (Eq. 8), multiple linear regression using annual variations around the mean (Eq. 9) (both with the y-intercept set to 0), and log-linear regression (Eq. 10), which was recently proposed by Awasthi et al. (2024). Besides the regression coefficients, which represent the sensitivities, we can also calculate  $R^2$  to check how much variance is explained by the different regression models and to investigate whether the unexplained variance shows any patterns in relation to the sensitivity estimates.

- Besides deciding on the equation to fit (e.g., single or multiple regression), we need to decide on the fitting method when applying regression analysis (e.g., ordinary least squares or more advanced methods). Earlier studies found that the fitting method has little influence on sensitivity estimates (Andréassian et al., 2016). We also tested different methods such as lasso or ridge regression (e.g., Dormann et al., 2013) and performed multiple regression sequentially (so-called stepwise partial regression), but found this to lead to negligible changes. We thus do not further investigate the influence of the fitting method.
- All methods are applied to annually averaged values. Since averaging of variables over more than a year has also been suggested in the literature (Andréassian et al., 2016; Zhang et al., 2022), we briefly tested this approach but found that it did not lead to improvements or additional insights (for neither the theoretical nor the observation-based analysis).

Table 1: Methods used to empirically estimate streamflow sensitivity to precipitation and potential evaporation.

| Name                   | Short name     | Equation                                                                                            | Comments                                                                                                  | References                       |
|------------------------|----------------|-----------------------------------------------------------------------------------------------------|-----------------------------------------------------------------------------------------------------------|----------------------------------|
| Non-parametric         | Nonpara.       | $\frac{\text{median}(\Delta Q)}{\text{median}(\Delta P)}$ and $\frac{\text{median}}{\text{median}}$ | combined attacks of D                                                                                     | Sankarasubramanian et al. (2001) |
| Single regression      | Single Reg.    | $\Delta Q = s_P \Delta P$ and $\Delta Q = s_P \Delta Q$                                             | Cannot account for combined effects of $P$ and $E_p$ ; same as $Q = s_p P + c$                            | Andréassian et al. (2016)        |
| Multiple regression #1 | Mult. Reg. #1  | $Q = s_P P + s_{PET} PET(8)$                                                                        | Intercept is set to 0                                                                                     | Introduced here                  |
| Multiple regression #2 | Mult. Reg. #2  | $\Delta Q = s_P \Delta P + s_{PET} \Delta PET ($                                                    | Same as $Q = s_P P + s_{PET} PET + c$                                                                     | Andréassian et al. (2016)        |
| Log-log regression     | Mult. Reg. Log | $\ln(Q) = e_{P} \ln(P) + e_{PET} \ln(P)$                                                            | $e_P$ and $e_{EET}$ need to be rescaled by $\bar{Q}/\bar{P}$ and $\bar{Q}/\bar{P}ET$ to get sensitivities | Awasthi et al. (2024)            |

## 120 2.3 Analytical model of streamflow sensitivity and synthetic dataset

The Turc-Mezentsev model is a Budyko-type model that takes long-term average precipitation P, potential evaporation  $E_p$ , and a shape parameter n as input, and returns streamflow Q (and, complementary to that, actual evaporation  $E_a$ ). The parameter n is often set between 2 and 3 (see e.g., Lebecherel et al., 2013). Here we use a value of n = 2.5, was has been used in previous studies (Andréassian et al., 2016) and fits well with the observational data used here (observational data are described in Section 3; for a Budyko-type plot see Figure S1 in the Supplementary Information). We would like to stress that our focus here is not on obtaining a perfect fit, so the choice of n is not of major importance.

The Turc-Mezentsev model can also be used to analytically calculate the sensitivity of Q to P and  $E_p$  (which may be summarised to the aridity index  $E_p/P$ ) by calculating partial derivates, so that we obtain a single sensitivity curve, which is a function of the aridity index. The resulting curves for Q/P (as a reference),  $s_P$ , and  $s_{Ep}$  are shown in Figure 1.

$$Q/P = 1 - \frac{\left(P^{-n} + E_p^{-n}\right)^{-\frac{1}{n}}}{P} \tag{11}$$

$$\frac{\partial Q}{\partial P} = s_P = 1 - \left(1 + \left(\frac{P}{E_p}\right)^n\right)^{\frac{n+1}{n}} \tag{12}$$

$$\frac{\partial Q}{\partial E_p} = s_{Ep} = -\left(1 + \left(\frac{E_p}{P}\right)^n\right)^{-\frac{n+1}{n}} \tag{13}$$

**Figure 1:** (a) Streamflow fraction of precipitation (shown as 1 - Q/P), (b) streamflow sensitivity to precipitation  $s_P$ , and (c) streamflow sensitivity to potential evaporation  $s_{Ep}$  as a function of the aridity index  $(E_p/P)$ , all based on the Turc-Mezentsev model (Eqs. 11-13). The grey lines in panel (a) indicate the water and energy limit, respectively.

Equation 11 is – similar to other Budyko-type equations – usually applied at long time scales, for which changes in storage are negligible (so that  $P = Q + E_a$ ). Accordingly, also the sensitivities (Eqs. 12 and 13) should be seen as sensitivities to long-term changes P and  $E_p$ . Here, we relax this assumption and use these equations to study year-to-year variability. We note that the same assumption is (implicitly) made in many sensitivity studies, which use year-to-year variability to estimate how catchments respond to long-term change, even though there is evidence that year-to-year variability in Q also responds to changes in, for instance, storage (Tang et al., 2020). For the theoretical analysis carried out here, this does not matter, as we test how well different methods can estimate sensitivities from synthetically generated data with the assumption that storage changes are zero. It will, however, matter when interpreting sensitivity estimates based on observational data.

Based on the Turc-Mezentsev model, we create a synthetic dataset for which the actual sensitivities are known, which will serve as a baseline for the theoretical analysis that will follow. To do so, we first generate synthetic catchments with fixed long-term values of P and  $E_p$  that span a wide aridity gradient (both P and  $E_p$  range from 300 to 3000 mm/y). For each synthetic catchment, we then create 50 annual value pairs by sampling from a bivariate normal distribution with the standard deviations of P and P0 set to 15% and 5% of their means, respectively, and with a Pearson correlation P1 between P2 and P2 set to three different values: -0.5, 0, and 0.5. The standard deviations are chosen to be similar to the ones found in the observational dataset used (17% for P2 and 4% for P2; data are described in Section 3). The correlation is added because P2 and P3 are regularly


correlated in empirical data ( $\rho_P$  averages -0.42 for the catchments used here) and this is known to affect sensitivity estimates (e.g., Chiew et al., 2014). Note that we use Pearson correlation  $\rho_P$  for the generation of correlated data, but in all other cases we use the more robust Spearman rank correlation  $\rho_S$  to quantify the strength of variable associations. Having created synthetic P and  $E_p$  data, we use the Turc-Mezentsev model to calculate Q for each annual pair of P and  $E_p$  (Eq. 11) as well as the sensitivities corresponding to the long-term values of P and  $E_p$  (Eqs. 12 and 13).

In addition, we investigate the influence of data uncertainty, which affects every hydrological variable (McMillan et al., 2012, 2018). To do so, we added normally distributed noise to P,  $E_p$ , and Q (after they have been generated). We use 2.5% of each variable's mean as standard deviation, so that about 95% of the data will fall within  $\pm 5\%$  of the mean ( $\pm 2$  times the standard deviation). While observational uncertainty is often estimated to be around 10% or higher, especially for precipitation (McMillan et al., 2012, 2018), these values are not directly comparable, as data uncertainty is often systematic (e.g., due to precipitation undercatch) and might be partly averaged out when using annual sums. Since the main aim here is to investigate the potential impact of data uncertainty, we decided to use a standard deviation of 2.5%, bearing in mind that the actual impact may vary depending on the actual uncertainty of the different variables.

Overall, we obtain 6 synthetic scenarios: three different strengths of correlation, each without and with random noise. The different sensitivity estimation methods (Table 1) are then used to derive sensitivities  $s_{est}$ , which are compared to the actual sensitivities  $s_{act}$  (Eqs. 12 and 13). This is done both visually and by quantifying the relative error as  $e_{rel} = (s_{est} - s_{act})/s_{act}$  [%], which is then averaged across all samples by taking the arithmetic mean.

#### 2.4 Temporal analysis

As can be seen from Equations 12 and 13, the streamflow sensitivities themselves depend on the aridity index, so that a change in aridity will lead to a change in the sensitivities. We can therefore also use the analytical model to check how typical temporal trends found for *P* and *E*<sub>p</sub> translate into trends in sensitivities (or elasticities). To do so, we calculate sensitivities over 20-year moving blocks in a longer time series (at least 50 years), resulting in time series of at least 30 years of sensitivity estimates. To calculate trends of sensitivities over time, we use the Theil-Sen trend slope estimator (Sen, 1968). In order to calculate relative trends (in %), we normalize the trends by the sensitivities calculated in the first 20-year block.

## 3 Data



In addition to synthetic data, we also test different sensitivity estimation methods using observational data. Using different large sample datasets, we select catchments according to the following criteria: we only keep time series with less than 5% missing values, at least 30 years in length, a fraction of precipitation falling as snow < 0.2, and where Q does not exceed P. For the analysis, all the time series are aggregated to annual values based on the water year. Details are reported in Table 2.

We always use national forcing products to ensure high quality forcing data, as global products were reported to be associated with substantial uncertainties (Clerc-Schwarzenbach et al., 2024) for CAMELS-US and CAMELS-GB. We also made a brief comparison for CAMELS-AUS and CAMELS-DE. When comparing SILO and AGCD forcing data for Australia (both national products), the sensitivities remain similar (Spearman rank correlation  $\rho_S = 0.99$  and 0.97, mean absolute difference = 0.03 and 0.02, respectively). When comparing DWD-HYRAS and Caravan-based ERA5 (Kratzert et al., 2023) forcing data for Germany, the sensitivities change slightly ( $\rho_S = 0.94$  and 0.86, mean absolute difference = 0.08 and 0.09, respectively). We will come back to the issue of reliably estimating  $E_p$  in the discussion, but note that this is not the main focus here.

For the trend analysis, we focus on CAMELS-DE and CAMELS-AUS due to availability of long time series and only kept catchments with at least 50 years of data.

**Table 2**: Datasets used in the study and some of their characteristics. WY denotes the start of the water year (for Germany, October is used instead of November for consistency).

| Dataset        | Country          | WY      | Forcing products                                                      | Subset                                                                                            | # total | # temporal | References                                    |
|----------------|------------------|---------|-----------------------------------------------------------------------|---------------------------------------------------------------------------------------------------|---------|------------|-----------------------------------------------|
| CAMELS<br>-US  | USA              | October | $P$ : Daymet $E_p$ : Daymet (calibrated Priestley-Taylor equation)    | Entire dataset (reference gauges; see Newman et al. 2015)                                         | 482     | -          | Addor et al. (2017b);<br>Newman et al. (2015) |
| CAMELS<br>-GB  | Great<br>Britain | October | <i>P</i> : CEH-GEAR <i>E<sub>p</sub></i> : CHESS-PE (Penman-Monteith) | UK benchmark dataset (Harrigan et al., 2018)                                                      | 119     | -          | Coxon et al. (2020a)                          |
| CAMELS<br>-AUS | Australia        | July    | $P$ : AGCD $E_p$ : SILO (Morton wetenvironment)                       | Entire dataset (hydrological reference stations), additionally filtered based on "river di < 0.2" | 355     | 209        | Fowler et al. (2024)                          |
| CAMELS<br>-DE  | Germany          | October | $P$ : DWD-HYRAS $E_p$ : DWD-HYRAS (Hargreaves)                        | Based on reference stations included in ROBIN (Turner et al., 2025)                               | 165     | 144        | Loritz et al. (2024)                          |
| Total          | -                |         | -                                                                     | -                                                                                                 | 1121    | 353        | -                                             |

## 4 Results




## 4.1 Comparison of sensitivity estimation methods using an analytical model

The analytical Turc-Mezentsev model allows us to test the different sensitivity estimation methods (Table 1) and compare them to theoretical values, thereby illustrating how the methods perform under known conditions. Figures 2 and 3 show the resulting sensitivity estimates for all methods as a function of the aridity index. Table 3 lists the relative errors, which are also visualised as bar plot in Figure S2 in the Supplementary Information.

Overall, the estimation of streamflow sensitivities to potential evaporation  $s_{Ep}$  results in much larger relative errors than for sensitivities to precipitation  $s_P$ . The smallest relative errors are obtained in absence of noise and with no correlation between P and  $E_p$  ( $e_{rel}$  from -2 to 6%). In presence of noise, the performance degrades for most methods, especially for  $s_{Ep}$ . For  $s_P$ , multiple regression methods #1 and #2 lead to the lowest relative errors overall (average absolute  $e_{rel} = 2\%$ ), followed by log-regression (5%). For  $s_{Ep}$ , multiple regression method #1 leads to the lowest relative error overall (average absolute  $e_{rel} = 6\%$ ), followed by method #2 and log-regression (both 12%).

When P and  $E_p$  are correlated, the nonparametric method and single regression (i.e., methods that do not account for both P and  $E_p$ ) lead to large and systematic errors. For instance, with negative correlations of -0.5, streamflow sensitivities to precipitation ( $e_{rel}$  from 6 to 10%) and to potential evaporation ( $e_{rel}$  from 231 to 314%) are – on average – systematically overestimated (Table 3). By contrast, when correlations are set to +0.5,  $s_P$  ( $e_{rel}$  from -12 to -10 %) and  $s_{Ep}$  ( $e_{rel}$  from -311 to -266%) are systematically underestimated. For multivariate methods, the relative errors are much smaller, but we still find systematic errors, especially for  $s_{Ep}$  and in presence of noise. For instance, with negative correlations of -0.5 and with noise, method #1 underestimates  $s_{Ep}$  by -7%, while method #2 and log-regression both underestimate it by -14% (see also Figure 3). Yet even when P and  $E_p$  are not correlated, we find systematic underestimation of  $s_{Ep}$  ( $e_{rel}$  from -9 to -19%).

**Table 3**: Relative errors  $e_{rel}$  [%], rounded to full percentages, and absolute average of the relative errors for the different estimation methods. Figure S2 in the Supplementary Information also shows the values as bar plots.

| Variable/<br>Method | $ \rho_P = -0.5 $ No noise | $ \rho_P = 0.0 $ No noise | $ \rho_P = +0.5 $ No noise | $ \rho_P = -0.5 $ With noise | $ \rho_P = 0.0 $ With noise | $ \rho_P = +0.5 $ With noise            | Average (absolute) |
|---------------------|----------------------------|---------------------------|----------------------------|------------------------------|-----------------------------|-----------------------------------------|--------------------|
| SP                  |                            |                           | - 1.0                      |                              |                             | .,,,,,,,,,,,,,,,,,,,,,,,,,,,,,,,,,,,,,, | (00000000)         |
| Nonpara.            | 9                          | 0                         | -10                        | 6                            | -3                          | -12                                     | 7                  |
| Single Reg.         | 10                         | 0                         | -10                        | 7                            | -3                          | -12                                     | 7                  |
| Mult. Reg. #1       | 1                          | 0                         | 0                          | -2                           | -3                          | -6                                      | 2                  |
| Mult. Reg. #2       | 0                          | 0                         | 0                          | -1                           | -3                          | -6                                      | 2                  |
| Mult. Reg. Log      | 8                          | 6                         | 4                          | 6                            | 3                           | -1                                      | 5                  |
| SEp                 |                            |                           |                            |                              |                             |                                         |                    |
| Nonpara.            | 313                        | 4                         | -311                       | 232                          | -18                         | -267                                    | 191                |
| Single Reg.         | 314                        | 2                         | -309                       | 231                          | -19                         | -266                                    | 190                |
| Mult. Reg. #1       | -1                         | -2                        | -2                         | -7                           | -9                          | -13                                     | 6                  |
| Mult. Reg. #2       | 3                          | 2                         | 1                          | -14                          | -19                         | -33                                     | 12                 |
| Mult. Reg. Log      | 3                          | 5                         | 5                          | -14                          | -16                         | -29                                     | 12                 |

**Figure 2:** Streamflow sensitivity to precipitation as a function of the aridity index. Theoretical values are based on the Turc-Mezentsev model, fed with precipitation and potential evaporation values that exhibit different degrees of correlation and noise to simulate real observations. The estimates resulting from the different methods are shown as point clouds with a LOESS regression (the fraction of data points which influence the smoothing at each value is set to 0.1) to aid visualization.


**Figure 3:** Streamflow sensitivity to potential evaporation as a function of the aridity index. Theoretical values are based on the Turc-Mezentsev model, fed with precipitation and potential evaporation values that exhibit different degrees of correlation and noise to simulate real observations. The estimates resulting from the different methods are shown as point clouds with a LOESS regression (the fraction of data points which influence the smoothing at each value is set to 0.1) to aid visualization.

#### 4.2 Comparison of sensitivity estimation methods using observational data

Since the Turc-Mezentsev model is a very simplified representation of reality, methods that perform well compared to the theoretical values may not perform well based on observational data that are influenced by more than just P and  $E_p$  (e.g., storage changes from year to year) and that are subject to other types of uncertainty (e.g., systematic bias). Hence, we now apply two selected methods to a large sample of near-natural catchments. We decided to focus only on methods #1 and #2, since log-regression leads to very similar results as method #2 and all univariate methods lead to poor performance if P and  $E_p$  are correlated (see Figures 2 and 3). This is indeed the case for most catchments, with Pearson correlation  $\rho_P$  between P and  $E_p$  in the observational data being mostly negative and averaging -0.42 (see Figure S3 in the Supplementary Information).



When applying methods #1 and #2 to observational data, we find good agreement between the two methods for  $s_P$  (Spearman rank correlation  $\rho_S = 0.96$ ) and little agreement for  $s_{Ep}$  ( $\rho_S = 0.07$ ), shown in Figure 4. In particular, we find that 52% of values for  $s_{Ep}$  are positive when using method #2, while it is only 3% when using method #1. Theoretically, we would expect  $s_{Ep}$  to be negative, as an increase in evaporative demand should be related to a decrease in streamflow. In terms of explained variance by the multiple regression methods, the two methods perform similar. The overall median  $R^2$  is 0.65 for #1 and 0.68 for #2, indicating an acceptable fit but also substantial variation left unexplained.

**Figure 4:** Comparison of streamflow sensitivity to (a) precipitation ( $\rho_S = 0.96$ ) and (b) potential evaporation ( $\rho_S = 0.07$ ) calculated using multiple regression methods #1 and #2 with observations from 1121 catchments. The grey dashed line shows the 1:1 line.

Since there is little difference for  $s_P$  and method #1 provides the most realistic values for  $s_{Ep}$ , we now compare the empirically estimated sensitivities from method #1 to the Turc-Mezentsev model, shown in Figure 5. Overall, both  $s_P$  and  $s_{Ep}$  follow the theoretical pattern and decrease with increasing aridity. However, the theoretical values tend to be underestimated and there is a larger spread for  $s_{Ep}$ . We also note that  $R^2$  tends to be smaller for catchments further away from the theoretical curves (see Figure S4 in the Supplementary Information). The results are similar for elasticities (see Figure S5).

Another way to visualise the resulting sensitivities is to plot  $s_P$  and  $s_{Ep}$  against each other, which is shown in Figure 6 with each catchment coloured according to its aridity index. We can see that the Turc-Mezentsev model leads to a single curve, meaning that each  $s_P$  is associated with a unique  $s_{Ep}$ , both being a function of the aridity index (cf. Eqs. 12 and 13). The same holds true for elasticities  $e_P$  and  $e_{Ep}$ , which plot as a straight line that follows the so-called complementary relationship ( $e_P + e_{Ep} = 1$ ). The empirical patterns roughly follow the analytical ones in the case of sensitivities, and almost show a perfect match for elasticities. Note, though, that this refers to the overall pattern and not individual catchments, which may sit on the theoretical line but at the wrong location with respect to their aridity index.

**Figure 5:** Streamflow sensitivity to precipitation (a) and potential evaporation (b) calculated using multiple regression method #1 with observations from 1121 catchments. Both panels show empirically calculated values (dots) and theoretical values based on the Turc-Mezentsev model (solid lines). Note that the y-axes are capped for better visibility and that two catchments plot above 0.5 for  $s_{Ep}$ .

**Figure 6:** (a) Streamflow sensitivity to precipitation plotted against streamflow sensitivity to potential evaporation. (b) Streamflow elasticity to precipitation plotted against streamflow elasticity to potential evaporation. Both panels show empirically calculated values (dots in the back) and theoretical values based on the Turc-Mezentsev model (line in front), coloured according to the aridity index. The grey dashed line starts at the origin and has a slope of -1, so that values plotting above it imply that  $s_P > s_{E_P}$  (a) and  $e_P > e_{E_P}$  (b).



## 265 4.3 Change of sensitivities over time

The strong relationship between streamflow sensitivity and aridity index found when comparing many catchments (Figure 5) suggests that a change in aridity index over time will also lead to changes in sensitivities for individual catchments. Theoretically, the Turc-Mezentsev model predicts a decrease (in absolute terms) in both sensitivities as aridity increases. Using sufficiently long observational records, we can calculate actual trends using sensitivity estimates over different time blocks (here based on method #1) and compare them to theoretical trends (via Eqs. 12 and 13) based on observed changes in precipitation and potential evaporation (and thus aridity index).

The resulting trends are shown in Table 4 and Figures 7 and 8. We find an increase in the aridity index over time, especially in Germany. Accordingly, the sensitivities decrease (in absolute terms) over time in all cases. However, the trend magnitudes are stronger in observational data (between -26% and -70%) than in the analytical model (between -6% and -22%). In Germany, we generally find lower sensitivities than based on Turc-Mezentsev, but the trends in the observational data are larger (-26% vs. -6% for  $s_P$  and -70% vs. -11% for  $s_{E_P}$ ). In Australia, both the sensitivities and the trends are relatively close to the analytical model (-27% vs. -15% for  $s_P$  and -35% vs. -22% for  $s_{E_P}$ ), except for  $s_P$  at the beginning of the time period (around 1980).

In absolute terms, most of the observed trends are in the order of around 0.15, meaning that at the end of the 50-year period a catchment that originally experienced a decrease of 0.5 mm in Q per mm decrease in P would now experience a decrease of only 0.35 mm in Q per mm decrease in P. Conversely, an increase in  $E_p$  at the end of the time period would lead to a smaller absolute reduction in Q, even though  $E_p$  related trends are likely less reliable given the uncertainty discussed previously.

Table 4: Absolute and relative trends of streamflow sensitivities. Relative trends are normalised with the value from the first year.

|           | Empirical [-/50y] | Analytical [-/50y] | Empirical [%/50y] | Analytical [%/50y] |
|-----------|-------------------|--------------------|-------------------|--------------------|
| Germany   |                   |                    |                   |                    |
| SP        | -0.16             | -0.05              | -26               | -6                 |
| $S_{Ep}$  | +0.16             | +0.07              | -70               | -11                |
| Australia |                   |                    |                   |                    |
| $S_P$     | -0.15             | -0.07              | -27               | -15                |
| $S_{Ep}$  | +0.08             | +0.05              | -35               | -22                |

**Figure 7:** Change of streamflow sensitivities and other variables over time for 144 catchments in Germany. Shaded areas indicate the 25<sup>th</sup> and 75<sup>th</sup> percentiles and thick lines indicate the median of all catchments. Dashed lines indicate the trends calculated with the Turc-Mezentsev model for the sensitivities based on observed P and  $E_p$  data. Sensitivities are calculated over 20-year blocks with the middle year shown (e.g., 1980 indicates a block from 1970 to 1990).

**Figure 8:** Change of streamflow sensitivities and other variables over time for 209 catchments in Australia. Shaded areas indicate the  $25^{th}$  and  $75^{th}$  percentiles and thick lines indicate the median of all catchments. Dashed lines indicate the trends calculated with the Turc-Mezentsev model for the sensitivities based on observed P and  $E_p$  data. Sensitivities are calculated over 20-year blocks with the middle year shown (e.g., 1980 indicates a block from 1970 to 1990).

## 5 Discussion


#### 5.1 Comparison of sensitivity estimation methods using an analytical model

Overall, univariate methods are unreliable in presence of correlation between P and  $E_p$ , which is the case for most catchments studied here (average  $\rho_P = -0.42$ ). While the limitation of univariate methods has been reported before (e.g., Andréassian et al., 2016), our results explicitly link errors in univariate methods to the correlation between P and  $E_p$ . In addition, even multivariate methods show systematic deviations from the analytical curves. While the multivariate methods always lead to an underestimation of  $s_{Ep}$  (i.e., less negative values) in presence of noise, the curves shown in Figure 3 still tend to be more negative on average for  $\rho_P = -0.5$  and more positive on average for  $\rho_P = -0.5$ , which is the same general pattern as for the univariate methods. It is sometimes argued that multiple regression accounts for the correlation between predictors, yet this statement may be a bit misleading. Multiple regression estimates the effect of one predictor while holding the others constant, but strong correlations between predictors (multicollinearity) can inflate standard errors and produce poorly conditioned coefficients that are highly sensitive to small changes in the data (Dormann et al., 2013). In presence of strong correlations (e.g., when wet years are typically associated with reduced potential evaporation), there is little variation in one predictor that does not overlap with the other, making it difficult to estimate unique effects.

Multivariate methods perform relatively reliable for  $s_P$  (2-5% relative error on average), but they are less reliable for  $s_{Ep}$  (6-12% relative error on average). The absolute values of  $s_{Ep}$  are always smaller than for  $s_P$  for a given aridity index (see e.g., Figure 7a, where all theoretical values plot above the dashed line) and so is their year-to-year variability (standard deviation was set to 15% for P and 5% for  $E_p$ ), which could explain why  $s_{Ep}$  is generally more difficult to estimate. This is substantiated by two additional checks (not shown here): if we increase the noise (smaller signal-to-noise ratio), the relative errors for both  $s_P$  and  $s_{Ep}$  increase; and if we increase the year-to-year variability of  $E_p$  in comparison to P, the relative errors for  $s_P$  and  $s_{Ep}$  become higher and lower, respectively.

Overall, the general underestimation (in absolute terms) of  $s_{Ep}$  in presence of noise might thus largely be due to relatively little variation in  $E_p$  compared to P and to noise (cf. regression dilution), with additional bias due to correlation between P and  $E_p$ . While there are other regression methods that could be tested, our results suggest that the problem lies not primarily in the fitting method, but rather in general limitations of using (multiple) regression to estimate sensitivities from noisy and correlated data. In summary, despite the simplicity of our synthetic experiment, it illustrates that none of the methods can reliably estimate the sensitivities in all cases, suggesting similar or larger uncertainties for observational data.







## 325 5.2 Comparison of sensitivity estimation methods using observational data

## 5.2.1 Uncertainty in empirical sensitivity estimates

Overall, multiple regression methods #1 and #2 lead to similar values for  $s_P$ , but show large disagreement for  $s_{Ep}$ . This generally agrees with the results using the analytical model, which also showed larger disagreement for  $s_{Ep}$ . As discussed in Section 5.1, absolute values of  $s_{Ep}$  are always smaller than for  $s_P$ , and previous studies also reported that changes in precipitation dominate the streamflow response of catchments (Berghuijs et al., 2017; Zhang et al., 2023). In addition, year-to-year variability in  $E_p$  (standard deviation of 4% on average) tends to be smaller than for P (17%), so that the already smaller  $s_{Ep}$  might be more difficult to constrain empirically, since the signal-to-noise ratio is relatively low (cf. Chiew et al., 2014). Alternative options which may help to constrain sensitivity values through regionalisation are pooled regression methods (e.g., panel regression; Anderson et al., 2025; Awasthi et al., 2024). This may allow robust estimation at the regional scale, although at the expense of unreliable sensitivity estimates at individual locations.

More than half of the values based on method #2 are larger than zero, which would suggest an increase in streamflow with increasing  $E_p$ . This cannot be generally attributed to concurrent increases in P, because P and  $E_p$  are anti-correlated. A considerable fraction of positive or zero values for sensitivities (or elasticities) to potential evaporation or temperature was also reported in other papers (Anderson et al., 2022; Andréassian et al., 2016, 2025; Awasthi et al., 2024; Xiao et al., 2020; Zhang et al., 2023). While this may be perceivable in certain circumstances (e.g., when warmer years are associated with increased precipitation intensity, or due to melt water contributions) it is unrealistic for this to occur in more than half of all catchments, casting doubt on the reliability of these sensitivity estimates. Alternatively, if these sensitivities were to capture actual catchment behaviour, it would imply that both simple Budyko-type and more complex simulation models, which usually show negative sensitivities to  $E_p$  (e.g., Roderick & Farquhar, 2011; Xiao et al., 2020), omit or misrepresent crucial processes related to evaporation. While method #1 leads to values of  $s_{Ep}$  that mostly fall between -1 and 0, these values are often relatively small compared to the Turc-Mezentsev model. This might partly be a consequence of the method, which showed systematic underestimation in the synthetic experiment (Figure 3). Hence, these  $s_{Ep}$  estimates should also be interpreted with caution.

## 5.2.2 Patterns in empirical streamflow sensitivities and influence of catchment storage processes

Comparing observation-based sensitivities to the Turc-Mezentsev model, we find that both  $s_P$  and  $s_{Ep}$  tend to be lower in observational data. Since Turc-Mezentsev is a climate-only model, actual sensitivities to climate (P and  $E_p$ ) are lower in real catchments where other factors matter, too. These include storage processes, seasonal offset between P and  $E_p$ , snow, and various other factors that were shown to influence streamflow sensitivities (e.g., Andréassian et al., 2025; Weiler et al., 2025; Zhang et al., 2023). Still, especially  $s_P$  follows the theoretical pattern relatively well, substantiating the strong influence of the aridity index on streamflow sensitivities. Note that in less strictly filtered catchment datasets (see Section 2.5), the scatter is likely larger due to other influences, such as human interventions in the water cycle.






The catchments that fall further away from the Turc-Mezentsev curves tend to have lower  $R^2$  ( $\rho_S$  between sensitivity and relative deviation from the curve is 0.53 for  $s_P$  and 0.21 for  $s_{Ep}$ ), substantiating the idea that other predictors not included in the regression model matter, too. Though not the main focus of the paper, we briefly considered several other variables commonly hypothesised to influence annual Q variability apart from P and  $E_p$  (storage, seasonality, and snow fraction). In particular, we tested if the differences between the Turc-Mezentsev-based sensitivities and the empirically estimated sensitivities are correlated with any of these variables. We found a relatively strong correlation ( $\rho_S = -0.46$  for both  $s_P$  and  $s_{Ep}$ ) between the deviation from the Turc-Mezentsev curve and the baseflow index (baseflow divided by total streamflow, with baseflow estimated using a digital filter; UKIH, 1980), indicating the importance of storage processes. We thus also fitted a regression model that includes a storage term, here approximated by the average streamflow from the previous year (see Figure S6 in the Supplementary Information). This led to two main insights. First, the median  $R^2$  increased from 0.65 to 0.69, suggesting that the storage term adds some, but not much, explained variance. Second,  $s_P$  and  $s_{Ep}$  stayed almost the same, indicating that the storage term only explains additional variance not captured by the other sensitivities, but does not change their values. While additional variables can therefore be included in the sensitivity calculation, it is worth nothing that (unlike here) this could lead to changes in the sensitivity estimates. If the sensitivities depend on the regression model fitted, interpretation of the resulting regression coefficients becomes more challenging. Also, regression models with many predictors increase the risk of overfitting (e.g., to certain combinations of catchments with specific correlation structures), especially when predictors are correlated or the signal-to-noise ratio is low. Thus, while including additional variables may increase our understanding of the drivers of (annual) streamflow variations (e.g., Andréassian et al., 2025), it also necessitates a close look at the meaning and robustness of the resulting sensitivity estimates.

#### 5.2.3 The complementary relationship

Interestingly, method #1 appears to enforce the so-called complementary relationship (Awasthi et al., 2024; Dooge, 1992; Zhou et al., 2015). This might be due to the nature of the equation used. If we reformulate Eq. 8 and substitute the sensitivities with elasticities, we get:

$$Q = s_P P + s_{Ep} E_p = e_P \frac{\overline{Q}}{\overline{P}} P + e_{Ep} \frac{\overline{Q}}{\overline{E_p}} E_p$$

$$1 = e_P \frac{\overline{Q}}{\overline{P}} \frac{P}{Q} + e_{Ep} \frac{\overline{Q}}{\overline{E}_p} \frac{E_p}{Q} = e_P \frac{\overline{Q}}{\overline{P}} \frac{(\overline{P} + \Delta P)}{(\overline{Q} + \Delta Q)} + e_{Ep} \frac{\overline{Q}}{\overline{E}_p} \frac{(\overline{E}_p + \Delta E_p)}{(\overline{Q} + \Delta Q)}$$

If we assume that the fluctuations are much smaller than the means, the means cancel out and we get:  $1 \approx e_P + e_{Ep}$ .





While the complementary relationship should hold if catchments follow Budyko-type behaviour (i.e., Q is solely controlled by variability in P and  $E_p$ ), real catchments do not necessarily behave that way. As soon as other factors (e.g., storage, changes in vegetation, human impacts) strongly affect a catchment's water balance and/or its sensitivity, we should not expect the relationship to hold. It is worth noting that the complementary relationship may be extended to account for elasticities to any number of driving variables (cf. Zhou et al., 2015). Yet, this should only be valid if all (major) drivers of streamflow variability are accounted for and in absence of large uncertainties. Overall, the so-called complementary relationship can thus be used to constrain sensitivity estimates, but also invokes assumptions that should be stated and assessed.

#### 5.2.4 Other uncertainties and limitations

There are various sources of uncertainty when working with observational data, which will affect empirically estimated sensitivities. Besides measurement uncertainty, catchment averages of *P* and *E*<sub>p</sub> rely on spatial interpolation procedures that introduce uncertainty (McMillan et al., 2012, 2018). These uncertainties are typically not purely random and may affect empirically estimated sensitivities in systematic ways. For instance, systematic underestimation of *P* (especially its variability) will lead to lower sensitivities. But even random uncertainties can affect sensitivity estimated based on linear regression, since large uncertainties compared to natural variability (i.e., a low signal-to-noise ratio) will impact the accuracy of regression coefficients, as shown in the synthetic experiment.

To minimise uncertainties arising from observational data, we selected catchments that are relatively unimpacted and come with long, mostly complete time series. We also only used national forcing products, since they are usually less uncertain than global products (Clerc-Schwarzenbach et al., 2024). In the case of the German catchments, for instance, the Hargreaves  $E_p$  estimates contained in CAMELS-DE differ substantially from Penman-Monteith  $E_p$  estimates ( $\rho_S = 0.38$ ) contained in Caravan. Independent of whether we deem the national  $E_p$  estimates to be more realistic, the large differences suggest considerable uncertainty. More generally, since  $E_p$  is not a measurable but a modelled quantity, it is also associated with substantial conceptual uncertainty. One alternative option would be the use of net radiation (normalised by latent heat of vaporisation) directly to avoid the use of a model for estimating potential evaporation, which may become even more uncertain when looking at climate projections (cf. Milly & Dunne, 2016).

# 5.3 Change of sensitivities over time

When investigating long time spans (50 years), we find that – in accordance with the analytical model – the sensitivities decrease as aridity increases. This finding is mirrored in results previously reported for elasticities in the US (Anderson et al., 2025), though the trends there were less clearly expressed, differed between regions, and were not explicitly related to trends in aridity. While not directly comparable, the relative trends estimated here (decrease between -26% and -70%) are in a similar range as the relative variations in interannual elasticity compared to long-term elasticity estimates (between 4 and 48%) found

by Anderson et al. (2025), which also showed lower elasticity in hot and dry years. This suggests that temporal changes in sensitivities of that order of magnitude are a more widespread phenomenon.

Overall, both  $s_P$  and  $s_{Ep}$  decrease by more than 20% over 50 years, suggesting that sensitivities may not be robust metrics for mid- to long-term projections. This holds true even without considering other factors that might influence the water balance in a future with elevated  $CO_2$  concentrations (e.g., interactions between P and  $E_p$ , or changes in stomatal conductance). At the same time, the change in sensitivity may be roughly estimated and projections adjusted accordingly, especially if trends are clearly expressed in empirical observations, as is the case for the Australian catchments. For the German catchments, the decrease is larger than predicted by the analytical model and the sensitivities themselves are much lower than the Turc-Mezentsev estimates. This might be due to due to data issues (especially  $E_p$ ), or because these catchments are less climate-driven on average, for instance due to human impacts, changes in land cover, or larger influence of catchment storage.

#### 6 Conclusion





A systematic comparison of empirical, primarily regression-based, methods for estimating streamflow sensitivities to precipitation and potential evaporation indicates that streamflow sensitivity estimates are often highly uncertain, especially for potential evaporation. This applies both to a synthetic experiment using an analytical model (Turc-Mezentsev) and to observations from >1000 near-natural catchments, and can also be transferred to streamflow elasticities. While multivariate regression methods are preferable over univariate methods, the commonly employed multiple regression approach resulted in unrealistic streamflow sensitivities to potential evaporation for the majority of catchments studied here. Using a variant of multiple regression (with an intercept of zero) resulted in the lowest relative errors in the synthetic experiment and leads to fewer unrealistic values when applied to observational data, likely because it enforces the complementary relationship (which states that elasticities to P and  $E_p$  should sum up to 1). While this method therefore appears preferable, it invokes strong assumptions and may still underestimate (or poorly estimate) sensitivities to potential evaporation. One possible reason for the difficulty in estimating sensitivity to potential evaporation is the low year-to-year variability in potential evaporation compared to precipitation and compared to observational uncertainty. Using unrealistic estimates of sensitivities to potential evaporation may be particularly problematic in regions where climate change impacts are largely driven by changes (typically increases) in evaporative demand and not precipitation totals, as is the case for Germany. In such instances, sensitivity-based projections may be deemed unreliable even if sensitivity to precipitation is accurately estimated. In addition, both theoretical and empirical results show that sensitivities decrease over time as aridity increases, indicating that static sensitivities may be unreliable for projections of climate change impacts. These results should therefore urge caution in the use of empirical sensitivities for both short-term and long-term projections, highlighting the need for further investigation.

# Data and code availability

CAMELS-US is available at <a href="https://dx.doi.org/10.5065/D6MW2F4D">https://dx.doi.org/10.5065/D6G73C3Q</a> (Addor et al., 2017; Newman et al., 2014). CAMELS-GB is available at <a href="https://doi.org/10.5285/8344e4f3-d2ea-44f5-8afa-86d2987543a9">https://doi.org/10.5285/8344e4f3-d2ea-44f5-8afa-86d2987543a9</a> (Coxon et al., 2020b). CAMELS-AUS v2 is available at <a href="https://zenodo.org/records/14289037">https://zenodo.org/records/13837553</a> (Dolich et al., 2024). Caravan is available at <a href="https://zenodo.org/records/7944025">https://zenodo.org/records/7944025</a> (Kratzert et al., 2025). Python code to reproduce the results and figures can be accessed at <a href="https://github.com/SebastianGnann/Streamflow\_sensitivities">https://github.com/SebastianGnann/Streamflow\_sensitivities</a> and will be permanently archived upon acceptance of the manuscript. Code development was aided by Perplexity and GitHub Copilot.

#### **Author contribution**

SG conceived the study, refined it based on discussions with all co-authors, and performed all analyses. SG prepared the manuscript with contributions from all co-authors.

# **Competing Interests**

MW is a member of the editorial board of the journal Hydrology and Earth System Sciences.

## Acknowledgements

We thank Felix Radtke for providing insights on streamflow elasticities in Germany and Carsten Dormann for advice on multiple regression methods.

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
