# Peer review of "Uncertainty and non-stationarity of empirical streamflow sensitivities"

_EGUsphere, 2025_

## Author Comment (AC1)

**Comments in black**
**Replies in blue**

**Referee #1**

**SYNTHESIS**

This paper deals with the precipitation and potential evaporation sensitivity of streamflow. It presents a theoretical study on the impact of different uncertainty sources which is very original, and allows to discard definitively one of the classical methods to identify elasticity (never seen anywhere in the literature... would be worth a technical note in itself). Then the paper goes on to show that the ongoing climatic change has already changed the empirical precipitation elasticity of streamflow in Germany, a very interesting and original result in itself.

We thank the reviewer for the detailed assessment and the helpful comments.

**OVERALL COMMENT**

This is a very good paper: excellent substance, excellent analysis, excellent form.

I would like in particular to congratulate the authors for using the sensitivities / absolute elasticities which are easily and logically interpretable (and have easily identifiable physical limits) instead of the relative ones ('true' elasticities). The plots showing the dependency of the relative elasticities (derived from the Turc-Mezentsev formula) to aridity, published elsewhere in the literature may be mathematically right but is useless in hydrological terms (the behavior with aridity makes no sense: we, as hydrologists, are not interested to know that a theoretical ratio of two terms that tend towards zero has a mathematical limit, we are interested to know that the two terms tend towards zero).

As a reviewer, my only recommendation is "don't change a word and publish as it is".

Thank you!

But since I am not only a reviewer but also a hydrologist interested in the topic, I could not help to comment your paper below. Feel free to consider or not my suggestions. I realize that there is enough matter to publish several very interesting papers, and I am definitely not requesting you to turn this paper into a very long undigestible paper.

Honestly, my only regret is your title, which is a little vague and not at the level of your work. The fact is that there are several very interesting points in your paper, it may be difficult to choose one over the others. Also, I guess that a strict statistician would argue that the term non-stationarity is not well-chosen, and would prefer you to talk about changing behavior, I remember a discussion with Prof. Koutsoyannis 10 years ago on this topic (see e.g. Efstratiadis et al., 2015).

Thank you for bringing this up. We will replace the word non-stationarity with temporal variability. Regarding the rest of the title, we have so far not found a better alternative without making it overly long, but we will think about it again when revising our manuscript.

**DETAILED COMMENTS**

The introduction is excellent

Thank you!

Fig1: why did you choose to plot 1-Q/P and not Q/P? Also, I have a problem with your physical limits: when you use catchment data the water limit should correspond to Q=P and the energy limit to Q=P-EP. Your water limit only corresponds to the "physical limit" (Q=0). I agree that it will be strange to have a "water limit" at 0, this is why I would personally use prefer to plot Q/P and not 1-Q/P. See e.g. Fig. 1 in the paper by Andréassian et al. (2025).

This is a good point and we will change the figure in our revised manuscript.

Table 1: please adjust your notations (PET -> $E_P$) for homogeneity with the rest of the paper

Thanks for spotting this, we will fix that.

Table 1: I understand what you mean by "same as $Q = s_P P + s_{\mathrm{PET}} PET + c$" even if I do not completely agree: when expressed in deltas, the formula offers other opportunities, such as the pooled regression which you mention, and is interesting to reduce the uncertainties and yield hydrologically coherent values

What we meant is that for the present case, it practically makes no difference. Since the $c$ parameter "incorporates" the means, the fitted sensitivities are exactly the same (we also tested this to make sure). We will clarify that and change the phrasing.

Table 1: I do not understand why you introduced eq #1... a very (too?) simplistic choice, unless you want to show us that when the elasticity coefficients try to adjust to represent at the same time the intercept of the regression and the slope, strange things happen (do you really need to add this option in this paper?)

We added "Multiple Regression #1" because it leads to the "best" sensitivity estimates in the present case. We did not originally expect that and we agree that setting the intercept to 0 is a strong assumption. But since it leads to the lowest error (Table 3) and to the most realistic range of sensitivity values for potential evaporation (Figure 4b), we decided to keep it. In addition, we think that its link to the complementary relationship is interesting as it also highlights the strong assumptions inherent in that relationship (namely that the elasticities have to sum up to 1). We are happy to emphasize again that this is by no means a perfect choice, but it would feel a bit odd to completely omit it.

l130: does it make sense to assume positive correlation between delta Q et delta E? 0 and negative could be enough? Unless there are places with that type of correlation in Australia perhaps.

As can be seen from Figure S3 in the supplement, there are indeed only a few catchments with positive correlations (16 to be precise; located in the US, Great Britain, and Germany). But since there are some in most of the studied countries, we decided to keep it.

Also in the generation procedure, I note that all the catchments are considered conservative. However, in many datasets (made of mostly small catchments) catchments are significantly contributing to regional aquifers, they "leak". I am not sure it's worth to include this aspect in your theoretical experiment, but it could be worth to discuss potential specificities of "leaking" catchments in the discussion part (the fact is that the leaked quantity itself can be sensitive to Precipitation).

This is a good point. It is true that this could also be tested in our theoretical experiment and it would certainly be quite interesting to do so. One way to do so would be the inclusion of a systematic bias in addition to random error. For instance, we could reduce streamflow each year by 10% to mimic a leaky catchment. The effect should be relatively straightforward, though, at least if the reduction is 10% each year: losing catchments will show lower sensitivities (but the same elasticities) because the absolute streamflow anomalies show a smaller spread. This is similar to precipitation undercatch, where the opposite should happen, because precipitation variability will be underestimated.

So, since this happens in real catchments, it certainly also explains some of the variability compared to the Turc-Mezentsev model. From a practical point of view, the difficulty then lies in identifying places where this is actually the case, which is probably too much for the current manuscript.

We will therefore not add any additional analyses, but discuss this in more detail in our revised manuscript.

l170 Temporal trend: it is not very clear which type of trends you apply (I understood later that you just use the observed trends: it would be good to add a sentence here on this).

Thanks for pointing that out. We will expand the description to make it clearer.

Fig3: I understand now that the nonparametric method of Sankarasubramanian et al (2001) gives wrong (i.e. positive) sensitivities of streamflow to $E_P$. Did anybody in the literature ever mentioned that?

Good question. The method is mostly used for sensitivities to precipitation, since studies investigating sensitivities to potential evaporation are usually aware of the strong confounding effect of precipitation, thus using multiple regression or Budyko-type approaches. So, to our knowledge, it has not yet been reported but also not often been used. We are happy to emphasize that.

l240, Fig 4: it would be interesting to note that one of the methods respects (almost) the physical limits, while the other does not. But I still do not understand why you introduced the multiple Regression #1 with intercept set to 0. It was obviously a bad choice... the model uses the degree of freedom of $s_P$ (which is often not even statistically significant) to compensate for the lack of intercept in the regression: you end up fitting $Q = aP + b$ and not $Q = aP + bE_P$

We mention that one of the methods mostly respects the physical limits in l.236-237 of the manuscript, but we will make this clearer.

Regarding Multiple Regression #1: as discussed above, we agree that it is not a perfect choice, but we did want to present it as it yields interesting results. In addition, if we were to use the results for $s_{Ep}$ from Multiple Regression #2 in the rest of the manuscript, we would end up with poorly defined sensitivities to potential evaporation. This would make the follow-up analyses much less clear, as shown below, where we compare the trends in $s_{Ep}$ for Australia and Germany using both methods.

[Figure]

Given that MR#2 is the most commonly used method, we will add all results using MR#2 to the supplement, so that interested readers can also have a look at them.

Fig6: very interesting graph

l265: when you look at the change of sensitivities with aridity defined as a variable it is like using a second-order assumption of "space-for-time trading". You should mention it.

We will add this to our revised manuscript.

l274: this is a very interesting and very original result of your paper. I would be particularly interested to understand whether it is linked to a change in seasonality (not accounted for in the regressions), to the incapacity of the $E_P$ model to represent the true evolution of the evaporative demand of the atmosphere, or to some other factor... A suggestion (for another paper) would be to test a monthly or daily rainfall-runoff model. If it is able to represent the change of precipitation sensitivity, then the problem is due to the seasonality that the annual anomaly model cannot account for. Otherwise, it means some other hydrological process is unaccounted for (or the $E_P$ model is inappropriate).

This is an interesting question. When plotting the evolution of seasonality over time (using the seasonality index based on Woods, 2009), we find little overall changes in Australia (but a very wide range) and a slight increase in Germany, suggesting that rainfall has become slightly more summer dominant. Note that this index, which is based on how the seasonality of $P$ aligns with the seasonality of $T$, is 1 for summer dominated catchments and -1 for winter dominated ones.

[Figure]

If we assume that summer dominant (i.e., more in phase) rainfall leads to less streamflow, we would expect a greater reduction in $Q$ than predicted by the Turc-Mezentsev formula. This isn't the case, however, because when we compare observed $Q$ to predicted $Q$ we find an almost perfect match (note that in response to R#2 we now use a calibrated $n$ value for the theoretical results in this plot). The reduction in the sensitivities does match the increase in seasonality, though, so this certainly could be a reason. For now, we think it is best to view this as a hypothesis, which will require more detailed testing, possibly including a model as suggested or an analysis of sub-annual sensitivities (e.g., weekly; Weiler et al., 2025).

We will also discuss limitations of the $E_p$ model and other data issues as an alternative hypothesis, even though using different data sources does not fundamentally alter any of the patterns found here (see reply to R#2).

In addition, when relating sensitivity anomalies (i.e., differences between Turc-Mezentsev and empirical sensitivities) to catchment characteristics, we find the highest correlation with BFI and only a very weak correlation with seasonality. So, another hypothesis would be that slowly responding systems show lower sensitivities and lower trends because they are less directly driven by annual climate fluctuations. Perhaps this influence or other influences have become even stronger in recent years, explaining the stronger than expected reduction, though this remains speculative. This could be tested in future studies by also checking how storage sensitivities have changed, for instance using a lag-1 autocorrelation term (cf. supplementary Figure S6).

Long story short: there appears to be no simple explanation for the observed patterns and they likely originate from a mix of reasons. We will thus expand the discussion and add some additional plots to the supplement, but leave a detailed investigation for future work.

Table 4: I am not sure to understand the sign of the relative values (and the necessity for them, if it is complicated to understand). Is this table really useful? Fig 7 and 8 are already extremely clear.

We wanted to give some actual numbers and not just figures, but since R#2 also commented on this table, we will remove it. With regard to the sign: the sensitivities decrease, hence the negative sign. Since all our values decrease, we could also remove it, but it is technically more accurate.

Figure 7 and Figure 8 are extremely clear and interesting. Without requesting too much additional analysis (because it would turn your paper into a book...), I was wondering whether the Australian dataset would allow for producing 2 subsets one with P and $E_P$ out of phase (the Mediterranean part of Australia), and another with P and $E_P$ in phase: I believe this could help to interpret the trend observed in Germany. Another solution would be to compute an index characterizing the P-$E_P$ phase-shift.

Thank you for the suggestion. To follow up on this, we first calculated a seasonality index for all catchments. From the figure below, we can see a tendency for a weak summer maximum in Germany, whereas Australia can be divided into a summer and a winter dominant part.

[Figure]

We then had a look at the relationship between the seasonality index and the sensitivities. Overall, the influence of seasonality on both $Q$ and the sensitivities is neither very strong nor uni-directional in our dataset. We actually find the strongest seasonality-related deviations in some Australian catchments, which, however show higher sensitivities (and higher streamflow) despite being heavily summer dominant. This rather matches Potter et al. (2005) who relate this to infiltration excess runoff following summer storms.

[Figure]

We then also split up the Australian dataset into two subsets with seasonality indices below and above 0, which show quite different patterns. In summer dominant regions, the empirical sensitivities are somewhat larger, matching the figure above. In winter dominant regions, we find the opposite and generally stronger differences between empirical and theoretical trends. So, the previously well-matching pattern for Australia appears to be in part an artefact of averaging across various climates in which the Turc-Mezentsev model both over- and underestimates the sensitivities. Generally, if the theoretical model cannot reproduce average sensitivities well, it also cannot reproduce the trends well, hinting at a common underlying reason (e.g. processes omitted in the model).

[Figure]

While these patterns are interesting, they are somewhat counter to our expectations. In particular, the catchments in Australia with winter dominant rainfall (i.e., Mediterranean) show a similar behaviour as the German catchments, which have slightly summer dominant rainfall. So perhaps it is not seasonality but some other catchment characteristic that explains these patterns. One characteristic of the German dataset is that the BFIs are largest on average, so we also compared the BFIs for the two Australian subsets, shown below. We can see that the winter dominant subset has a higher BFI on average (it is also somewhat less water-limited; not shown here), thus being closer to the German catchments in terms of streamflow dynamics. While also not a conclusive explanation, it certainly represents an alternative hypothesis.

[Figure]

So, overall, we are left with many potentially interesting additional results, but need to restrict the amount of analyses presented to end up with a reasonably concise paper. Our suggestion therefore is to include the two Australian subsets in the trend analysis and add a brief discussion (but not too many additional analyses) on potential reasons for the different trends found in Germany and the two Australian subsets.

l336: you write "More than half of the values based on method #2 are larger than zero". But I guess that anyway the p-values from a Student t-test would consider these values as non-significantly different from 0. Perhaps mention it?

When using method #2, indeed many of the p-values for $s_{Ep}$ are smaller than 0.05 (if we chose this as a typical, but arbitrary, threshold). Interestingly, though, the values that are significant are both positive and negative. For method #1, many more values are significant, though of course that might in part be because the second term compensates for a lack of an intercept. We will add the p-values to the supplement and briefly discuss them in our revised manuscript.

[Figure]

**REFERENCES**

Andréassian, V., Guimarães, G.M., de Lavenne, A., and Lerat, J.: Time shift between precipitation and evaporation has more impact on annual streamflow variability than the elasticity of potential evaporation, Hydrol. Earth Syst. Sci., 29, 5477–5491, https://doi.org/10.5194/hess-29-5477-2025, 2025.

Efstratiadis, A., Nalbantis, I., and Koutsoyiannis, D., 2015. Hydrological modelling of temporally-varying catchments: facets of change and the value of information. Hydrological Sciences Journal, 60 (7–8). doi:10.1080/02626667.2014.982123

---

## Author Comment (AC3)

**Comments in black**
**Replies in blue**

**Referee #2**

Gnann et al. focus on the robustness and stationarity of streamflow sensitivities to P and Ep because the concepts of sensitivity and the methods used to estimate it are not fully clear in the literature. They approach this by: (1) generating six combinations of synthetic data from the Turc-Mezentsev model to identify methods that perform reliably across conditions; and (2) applying the selected methods to catchments with long-term observations to explore how sensitivities evolve over time and the sources of uncertainty. The manuscript is well structured and several of the results are very insightful. My detailed comments are below.

We thank the reviewer for their detailed and constructive review.

**Major comments:**

before going into the specific results, I think it would help if the manuscript clarified how the theoretical sensitivities relate to the empirical ones. The analytical sensitivies come directly from the Turc-Mezentsev curve, whereas the empirical values are estimated from interannual variability using regression. Because the Budyko curve is nonlinear, a regression slope over many years does not necessarily match the local derivative at the long-term mean. This difference might explain part of the mismatch between the analytical lines and the observations in several regions.

This is a good point. Indeed, the assumption behind the regression is that variability around the long-term mean is sufficiently linear. When looking at the Budyko curve we can see that this is approximately accurate if aridity only varies by a relatively small amount (e.g. 0.1 to 0.2, which for typical catchment averages may translate into a few 100 mm/y). In addition, some parts of the curve are more curved than others, likely making a linear fit less accurate.

The fact that we can recover the sensitivities very well (in absence of noise and correlation) shows that the regression approach generally works well in the range of variability used (here 15% and 5% standard deviation). We can also visually check the degree of (non-)linearity for some example cases, as shown below. We note that this bivariate view is somewhat limited, though, given that we actually look at a trivariate relationship here. We picked 4 catchments from the 4 countries investigated, ordered according to aridity, and 4 artificial cases that have a similar range in $P$ and $E_p$. We can see that the pattern is linear in most cases, but that weak nonlinear behaviour is visible for the more arid catchment (both in the observational data and in the artificial data). What is also visible from the figures, is that real catchments exhibit more scatter, resulting in lower sensitivities. This is particularly evident for the German catchment and matches the general pattern found here, namely that the German catchments show lower sensitivities on average.

We will add the figure to the supplement and discuss this issue more thoroughly in a revised manuscript. One option for future work would be to quantify the degree of nonlinearity and to check in which catchments such behaviour most likely occurs

related to the point above, using a single Budyko parameter n across all catchments can also affect the comparison. Since n controls the curvature of the Turc-Mezentsev relationship, regional differences in n would naturally show up as differences in the "expected" sensitivities, even though the manuscript notes that the exact value of n is not the focus. A fixed n can still influence the shape of the theoretical trends, so it would help to check how sensitive the analytical results are to this choice. This may be particularly relevant for Figs. 7 & 8, where the theoretical line captures the trend over Australia but not Germany. Labeling points by country in Fig. S1 might reveal if this mismatch is regionally systematic. The strong bias in German trends also make it difficult to interpret the degree of non-stationarity, even though this general pattern is consistent. Maybe consider to use boxplots or similar summaries to describe the catchment-level trends.

Thank you for bringing this up. One main reason why we do not want to focus on the parameter *n* is that it lumps together many, often co-related factors (e.g. related to storage, vegetation, seasonality, groundwater losses/gains etc.) and may even compensate for systematic data uncertainty (e.g. $E_p$ calculation, underestimation of *P*). So, independent of how it is assessed, interpreting the results will not be straightforward, especially not without an in-depth analysis.

We agree, however, that the mismatch can partly be attributed to the fact that the Turc-Mezentsev model does not capture local/regional conditions well. We thus calibrated the *n*

parameter (by minimizing the absolute error) for each national dataset to (a) get an idea how variable it is and (b) use this later on in the country-based trend analysis. For all other analyses (e.g. Figures 5, 6), we now use a calibrated value for all catchments (2.2). The figure below shows the catchments and the fitted curves. The lowest value is found for Germany (1.9), while the highest is found for Great Britain (2.7). We will add this to the supplement.

[Figure]

Using these *n* values for calculating the expected trends shifts them slightly, but the general mismatch remains, as shown below for the German catchments.

[Figure]

The main reason for the differences therefore seems to be a different one and is likely related to the fact that the Turc-Mezentsev does not capture the sensitivities and their temporal variability/trends well, even if it captures the mean water balance well. (As a side note, this implies that using the Turc-Mezentsev calibrated on many catchments does not predict well how individual catchments respond to changes in aridity.) We will discuss possible reasons for the differences between theoretical and empirical trends more thoroughly in a revised manuscript, see also our reply to R#1.

for the data used, the national forcing of P is very useful, but more information is needed on how Ep is calculated in each region. Why are different formulations used in these datasets? If these formulations were chosen because they best represent local conditions, it would be good to explicitly clarify. If not, part of the apparent non-stationarity in sensitivities might come from the way Ep is estimated. This might also help explain the positive s_Ep values in Fig. 4b. A brief comparison with an alternative Ep method (such as PET_Yang2019) might be benificial, although I think it may be somewhat beyond the main scope.

The reasons for using the different data products vary. Sometimes there just happens to be a national-scale product (e.g. Australia, Great Britain), sometimes the decision was made during the creation of the dataset, often as part of a modelling exercise (e.g. USA, Germany). We assume that the respective authors chose the $E_p$ estimation methods to work well for the national conditions. For example, in the CAMELS-DE paper it reads "This variant of the Hargreaves formula resulted in the lowest mass balance error in most catchments with respect to other methods (e.g. Penman, Priestly–Taylor) to estimate evapotranspiration […]." We will add some more details on the $E_p$ estimation methods used.

As briefly described in l.182-188, we compared sensitivities estimated using different $P$ and $E_p$ products for Germany and Australia, which were available in the respective datasets or as part of their Caravan extension. The scatter plots below show the resulting sensitivities using multiple regression #1. For Germany, for instance, the Spearman rank correlation between the national and the Caravan-based sensitivities are 0.94 for $s_P$ and 0.86 for $s_{Ep}$, and the mean absolute differences = 0.08 and 0.09, respectively. While this shows that there are some differences, it does not explain the apparent non-stationarity, nor the positive $s_{Ep}$ values. But we totally agree that $E_p$ is one of the most uncertain inputs here and will further emphasize this in our revised manuscript.

[Figure]

for the unexplained variation in sensitivities, it might be useful to discuss the role of vegetation. The vegetation cover influences the rainfall-runoff relationship, water storage; and the effective Budyko parameter n. Long-term changes in vegetation traits could shift catchments relative to the theoretical curve and influence sensitivities to both P and Ep. Nijzink and Schymanski (2022) provide an interesting example of how adjustments in vegetation influence the Budyko parameter n, and connecting this to your results might strengthen the interpretation.

Thank you for this useful reference. We will discuss the role of vegetation and other potential influences on the sensitivities more thoroughly in a revised manuscript.

**Minor comments:**

for Line 114 & 117, could you provide these results in supplementary materials?

Sure, we will add a comparison of sensitivities estimated using 5-year averages and sensitivities estimated using annual averages to the supplement.

Lasso and ridge regression lead to virtually the same values, so we do not include them here.

for Table 1, (a) Log-log regression should be log-linear regression and $e_{EET}$ should be e_Ep; (b) why do you use PET and Ep together?

Thanks for pointing that out. That was a mistake and should all read $E_p$. We will fix it. We will also change it to log-linear regression.

for the Turc-Mezentsev model, how is Ep calculated? It directly influence s_Ep, s_P and Ep/P.

There is no need to calculate $E_p$ for the theoretical experiment. Both the $P$ and $E_p$ values are chosen as described in Section 2.3.

When I first saw Table 4, I misunderstood the relative trend, i.e. positive s_Ep with a negative relative change. I think this table is unnecessary.

We will remove the table and add the trend values to the figure caption.

**Reference:**

1. Yang, Y., Roderick, M. L., Zhang, S., McVicar, T. R. & Donohue, R. J. Hydrologic implications of vegetation response to elevated CO2 in climate projections. Nature Clim Change 9, 44–48 (2019).
2. Nijzink, R. C. & Schymanski, S. J. Vegetation optimality explains the convergence of catchments on the Budyko curve. Hydrology and Earth System Sciences 26, 6289–6309 (2022).